# Immune Responses to Gene Editing by Viral and Non-Viral Delivery Vectors Used in Retinal Gene Therapy

**DOI:** 10.3390/pharmaceutics14091973

**Published:** 2022-09-19

**Authors:** Duohao Ren, Sylvain Fisson, Deniz Dalkara, Divya Ail

**Affiliations:** 1Sorbonne Université, INSERM, CNRS, Department of Therapeutics, Institut de la Vision, 75012 Paris, France; 2Université Paris-Saclay, Univ Evry, Inserm, Genethon, Integrare Research Unit UMR_S951, 91000 Evry-Courcouronnes, France; 3Institut de la Vision, INSERM UMR S968, 17 rue Moreau, 75012 Paris, France

**Keywords:** CRISPR–Cas9, Adeno-associated virus, immune responses, innate immune response, adaptive immune response, ocular gene therapy

## Abstract

Inherited retinal diseases (IRDs) are a leading cause of blindness in industrialized countries, and gene therapy is quickly becoming a viable option to treat this group of diseases. Gene replacement using a viral vector has been successfully applied and advanced to commercial use for a rare group of diseases. This, and the advances in gene editing, are paving the way for the emergence of a new generation of therapies that use CRISPR–Cas9 to edit mutated genes in situ. These CRISPR-based agents can be delivered to the retina as transgenes in a viral vector, unpackaged transgenes or as proteins or messenger RNA using non-viral vectors. Although the eye is considered to be an immune-privileged organ, studies in animals, as well as evidence from clinics, have concluded that ocular gene therapies elicit an immune response that can under certain circumstances result in inflammation. In this review, we evaluate studies that have reported on pre-existing immunity, and discuss both innate and adaptive immune responses with a specific focus on immune responses to gene editing, both with non-viral and viral delivery in the ocular space. Lastly, we discuss approaches to prevent and manage the immune responses to ensure safe and efficient gene editing in the retina.

## 1. Introduction

Inherited retinal degenerations (IRDs) are conditions of the retina caused by mutations in genes mostly expressed in the retinal pigment epithelium or photoreceptors. IRDs can result in severe visual impairment or complete vision loss, and are a social and economic burden [1]. There are ongoing efforts to develop therapies for IRDs that include pharmacotherapy, neuroprotection, gene therapy, optogenetic therapy, retinal prostheses and stem cell therapy. Many of these approaches have entered the clinical phase of development and some of them have translated into approved products. Currently, gene therapy is the most promising approach to treat IRDs caused by a single recessive gene defect [2]. Gene therapy refers to the use of genetic material to counteract an inherited or complex disease. If applied early in the disease, it can take the form of gene correction, gene replacement or augmentation, which have a common goal in replacing the mutant gene with a healthy copy either at the chromosomal site in situ or by providing an additional healthy copy that is maintained extra-chromosomally [3]. In the past decade, the Clustered Regularly Interspaced Short Palindromic Repeat (CRISPR)-Cas9 system has become one of the most powerful tools for precise gene editing and is being considered for retinal gene therapy applications [4,5]. While this holds immense therapeutic promise, it also presents some tool-specific challenges, including off-target editing and immune responses to the Cas9 protein and/or to the guide RNAs [4,6].

The therapeutic gene can be supplied to the eye by viral vector delivery or non-viral delivery. Recombinant versions of a common virus are generated by replacing the viral genome with the therapeutic gene of interest [7]. Many engineered viral vectors have been successfully used for gene therapy applications [8]. Among these, the Adeno-associated virus (AAV) is the vector of choice for the majority of retinal gene therapies, owing to their excellent transduction and safety profile. AAVs provide long-term transgene expression, induce a low host immune response, infect both dividing and quiescent cells and do not integrate into the host genome, thereby reducing the risk to patients [9]. However, the large size of IRD genes is currently a limiting factor for AAV-mediated gene augmentation as the AAV has a packaging limit of ~4.5 kb [10]. There are other viral vectors that have a larger packaging capacity, such as Lentiviruses (LVs), which can carry genes of up to 8 kb [11]; and Adenoviruses (Ads), which can reach a packaging capacity of 37 kb [12]. LVs are enveloped retroviruses containing a positive, single-stranded RNA genome, capable of infecting both dividing and non-dividing cells. Their major drawback is that their large size limits their diffusion and transduction capacity of the neural retina. Additionally, they are genome-integrating, and their integration can at times occur at an oncogenic locus [13]. Ads have a large packaging capacity, but they are highly immunogenic, thereby decreasing both the efficiency and safety of gene therapy [14,15,16].

Non-viral delivery is a broad-spectrum term used for many delivery methods that do not use viral packaging. These may include, but are not limited to, strategies that functionalize (by an addition of peptides or charges) proteins or DNA for direct delivery, use nanoparticles or use lipid-based coating. The intention of developing such techniques is to have larger packaging capacities or to deliver genes without packaging to avoid viral components—and hence, be less immunogenic—and for the ease of large-scale productions [17]. However, so far most of these strategies have reported low efficiency and could not accomplish transgene expression at therapeutic levels [18]. Additionally, as elaborated in the following sections of this review, immune responses induced by non-viral vectors have also been reported [19].

Most of the early advances in gene therapies were achieved in the eye, owing to the convenient tissue access and immune-privilege of the eye. In addition, this region also ensures the ability to monitor non-invasively, and ensures the existence of the contralateral eye as an in vivo control [8]. However, adverse outcomes from some clinical trials and evidence from animal studies have demonstrated that the immune-privilege of the eye is not absolute [20]. The introduction of foreign substances in the form of DNA, protein, chemical compounds and nanoparticles can elicit both innate and adaptive immune responses [17]. Innate immune responses induced by the vector, transgene or protein products can promote local inflammation in the eye, leading to deterioration of visual acuity. Innate immune responses can further boost adaptive immune responses to generate antibodies or cytotoxic T cells, which limit the transduction efficiency of the vector and/or clear the transduced cells (Figure 1) [21].

In this review, we first discuss studies that have reported on pre-existing immunity towards viral vectors and the Cas9 protein. We then evaluate both innate and adaptive immune responses with a specific focus on immune responses to gene editing, both with non-viral and viral delivery in the ocular space, and finally discuss approaches to prevent and manage the immune responses to ensure an increase in safety and efficacy of viral or non-viral vector-mediated gene editing in the retina.

## 2. Pre-Existing Immunity

Pre-existing immunity generally refers to the presence of antibodies in a host against components of therapy which can interfere with the therapeutic intervention. The most widely used Cas9 orthologs are derived from Staphylococcus aureus (SaCas9) and Streptococcus pyogenes (SpCas9) [4], which are common human commensals that could sometimes become pathogenic [22,23]. Therefore, it is reasonable to assume the presence of pre-existing immunity against Cas9 proteins in the human population that would affect the efficacy and safety of therapies involving Cas9 proteins. A study involving 125 healthy adult human blood donors in the USA with a median age of 43 reported pre-existing anti-SaCas9 antibodies and anti-SpCas9 antibodies in 78% and 58% of the total samples, respectively. In this study, apart from the antibodies, pre-existing cellular immune response was also evaluated. The authors reported an increase in the cells releasing IFN-γ in peripheral blood mononuclear cells (PMBS) from 18 donors stimulated with SaCas9 and SpCas9. Among the tested donors, 78% had SaCas9-specific T-cells and 67% were positive for SpCas9-specific T-cells. Cytokine positive cells were observed by Intracellular Cytokine Staining (ICS) targeting IFN-γ, TNF-α and IL-2. The presence of activated T-cells was reported by the FACS sorting T-cell activation markers—CD137 and CD154 [24]. Another study conducted on 48 healthy donors also confirmed this finding of a high prevalence of T-cells against SpCas9 protein, demonstrating that SpCas9 stimulation could activate CD137^+^ and CD154^+^ T-cells [25]. A study that compared the pre-existing anti-SaCas9 and anti-SpCas9 antibodies between serum and vitreous liquid from 13 patients who received vitreoretinal surgery with vitrectomy concluded that all the three serum samples were positive for SaCas9 and SpCas9, while only two vitreous fluid samples tested positive for SaCas9 and two others for SpCas9. This study also showed that the antibodies’ levels were higher in serum than in vitreous liquid [26].

An animal study conducted on 78 adult canines (60 WT and 18 models for Duchenne muscular dystrophy (DMD)) reported pre-existing anti-SpCas9 antibodies in all animals, but not in 16 newborn puppies, which only had moderate levels of maternally-derived Cas9 antibodies that dropped between two and six weeks of age. This study indicated that older animals in a study are more likely to be positive for anti-Cas9 antibodies than the younger ones [5].

AAVs are widely spread in the natural environment, and it has been reported that about 50–90% of the human population have been exposed to different AAV serotypes [27]. Exposure results in the presence of anti-AAV antibodies, which can further trigger a stronger response upon a second exposure, such as in the case of AAV-mediated gene therapy (Figure 1). A study conducted to evaluate pre-existing anti-AAV antibodies in 226 healthy donors between the ages of 25 and 64 years demonstrated that there is a prevalence of serum IgG to AAV in the healthy population. The highest seroprevalences were observed with AAV2 (72%) followed by AAV1 (67%), AAV9 (47%), AAV6 (46%), AAV5 (40%) and AAV8 (38%) [28]. Another study evaluated NAbs in 888 human serum samples from healthy volunteers from 10 countries. NAb assays were performed to detect antibodies against AAV1, AAV2, AAV7, AAV8 and AAVrh32.33 (a novel, structurally distinct AAV variant). The results showed that the highest prevalence was against AAV2, even though there was viability in samples from different continents. The anti-AAV1 and anti-AAV7 NAb levels were similar, followed by anti-AAV8, and not surprisingly, the lowest NAb titers were against the novel variant AAVrh32.33 [29]. In most human studies, the prevalence was highest against AAV2, while the lowest prevalence was against AAV8. However, a study conducted on 41 non-human primates (NHP) revealed that the highest level of antibodies was against AAV8 and AAV9 [30]. In humans as well as NHPs, the pre-existing antibodies against AAV5 tend to be lower [28,30]. Hence, there is pre-existing immunity against AAV among humans and animals used in research. However, their levels show variations by species, region and age. These findings are plausible, since the older the individual, the higher the probability is of them encountering the corresponding wild-type microorganism, thus, triggering the pre-existing adaptive immunity. Moreover, it is not surprising that the concentration of anti-Cas9 or anti-AAV immunoglobulins is lower in intraocular fluids due to the existence of ocular barriers, such as the blood-retinal barrier. However, a lower antibody concentration in the vitreous humor should not be underestimated because it is not yet clear whether a low level of Cas9 or AAV-specific antibodies can have a negative impact on the transduction efficiency and/or on the triggering of a local inflammatory response.

## 3. Immune Response to Cas Proteins

CRISPR-associated proteins (Cas) are present in most archaea and bacteria as adaptive immune systems [31] and provide sequence-specific resistance against phages [32]. A CRISPR–Cas system contains Cas proteins and small guide RNA sequences (sgRNA) that guide the Cas proteins to specific DNA binding sites that contain specific protospacer adjacent motif (PAM) sequences. This system has been adapted for a wide array of gene-editing applications, including gene therapy [33]. As an exogenous protein, Cas9 can be regarded as a foreign antigen in animals, and hence, can be presented by antigen-presenting cells (APC) to lymphocytes to induce host immune responses (Figure 1) [34].

The presence of ocular barriers allows the eye to limit inflammation, and therefore, makes it a good target for CRISPR/Cas9-based gene therapies [33]. A study aimed at comparing the serum and vitreous anti-Cas9 antibody levels in wild-type (WT) C57BL/6J mice immunized intramuscularly wild-type (WT) C57BL/6J mice with Cas9 intramuscularly, and reported a lower expression of antibodies in vitreous fluid compared to the serum, implying that preexisting immunity to Cas9 may present a lower risk in human eyes than systemically administered ones [26]. Angiogenesis is the process of the formation of new blood vessels from existing ones and is a hallmark of certain stages of eye diseases, such as wet age-related macular degeneration (AMD) [35]. A study aimed to use genome editing to treat angiogenesis-associated diseases by packaging Cas9 and gRNA targeting vascular endothelial growth factor receptor 2 (VEGFR2) in AAV1 and delivering it by intravitreal injections to the eye of WT C57BL/6J mice. They analyzed toxicity and inflammation by imaging the fundus as well as immunolabeling the retina for inflammation markers but reported no damage to the retina four weeks post-injection [36]. Mutations in the CEP290 gene caused a frequent form of LCA, and a CRISPR–Cas9-based gene therapy to correct this mutation called EDIT-101 was evaluated in a pre-clinical study. EDIT-101 showed high efficiency in editing the CEP290 gene in humanized CEP290 mice and WT non-human primates (NHP). This study also evaluated immune responses but reported no Cas9-specific antibodies in the serum, no activated T-cells in peripheral blood mononuclear cells (PBMCs) at 2, 4, 7 and 13 weeks post-injection and no obvious retinal damage was observed by fundus imaging [37]. This reiterates previous observations that there is probably a lower risk associated with CRISPR–Cas9-based therapies for the eye. The promising results of this pre-clinical study gave rise to a gene-editing product named EDIT-10, which uses AAV5 to deliver saCas9 with two guide RNAs to excise a region of the CEP290 gene bearing a splice acceptor site, resulting in a truncated gene product. However, there are no long-term results reported following the immune responses after the delivery of Cas9 in patients thus far.

Besides the Cas9 protein, the other component of the CRISPR/Cas9 system—gRNA—may also pose a risk of inducing an immune response. To explore this potential risk, guide RNAs with a 5′-triphosphate group (5′-ppp gRNAs) or gRNAs with 5′-hydroxyl were delivered to primary human CD4^+^ T cells with or without recombinant Cas9 as a ribonucleoprotein (RNP) complex. The level of IFN-β was measured by ELISA, while the expression of innate immune response-related genes—DDX58, IFNB1 and OAS2—was tested by qPCR. The results showed that there was a higher expression of innate response-related genes and IFN-β secretion in groups that received 5′-ppp gRNAs. It proved that gRNA might also trigger innate immune responses in murine and human cells, thus, leading to cytotoxicity. The authors conclude that chemically synthesized gRNAs with a 5′-hydroxyl group are more efficient and less immunogenic than in vitro–transcribed (IVT) sgRNAs in human and other mammalian cells. Thus, immune responses against the guide RNAs should also be considered to have a complete overview of the potential immune responses against the CRISPR/Cas9 system [38].

Although pre-existing immunity against Cas9 has been reported, the few studies evaluating post-injection immunity have not reported severe adverse effects in the eye. However, when Cas9 is delivered systemically, the response can be drastic. In a study aimed at evaluating Cas9-related immune responses, the Cas9 protein with GFP was delivered to the liver of FVB/NJ mice using an Ad vector. Fourteen days post-injection, CD4^+^ and CD8^+^ T-cell infiltration was observed in liver sections, while anti-Cas9 antibodies were found in the sera collected from the mice. Further, splenocytes harvested for the cytokine release assay after Cas9 protein stimulation resulted in a Cas9-specific cellular immune response. The transgene-GFP expression was observed on day 3, started decreasing from day 7 and finally disappeared at day 14 [39]. Another study aimed at evaluating transduction efficiency and host response, delivered split-spCas9 packaged into two AAVs by systemic and intramuscular injection in Ai928 and WT (C57BL/6) mice. Serums were collected to measure anti-SpCas9 antibody levels, and muscle tissue was harvested to test the immune cells’ profile by FACS. Interestingly, both AAV9 and Cas9 induced the production of specific antibodies, while only Cas9 induced the cellular immune response [40]. A study comprising of 42 dogs of three types of canine models with muscular dystrophy aimed at analyzing Cas9-specific immune responses locally and systemically. Animals received AAVs carrying Cas9 intramuscularly or intravenously. Both injection modes induced Cas9-specific immune responses, including humoral and cellular immune responses. CD8^+^ and CD4^+^ T-cell infiltration and Cas9 proteins were observed by the immunolabelling of muscle tissues after 3 weeks. Further, ELISpot assays revealed an increase in activated SpCas9-specific T cells, and the levels of different cytokines (IL-2, IL-15, IL-18, IFN-γ, TNF-α) were also elevated 6 weeks post-injection. Anti-SpCas9 ELISA performed on serum samples showed the production of anti-SpCas9 antibodies. The Cas9-edited cells in the muscles were cleared 6 weeks post-injection even though the animals were immunosuppressed by a high dose of prednisolone [5]. This finding indicates that in large animal models, anti-Cas9 immune responses remain a major obstacle that may limit the clinical applications of CRISPR/Cas9 systems, especially in non-immune-privileged tissues. These studies are also an indication that the response to the CRISPR–Cas9-based therapies can be severe; in addition, while such therapies are being designed for the ocular space, considerations should be given to the dose and pre-existing immunity, and the evaluation of immune responses should be carried out locally, i.e., by the evaluation of inflammatory biomarkers in the vitreous fluid and evaluation of cellular infiltration in the ocular tissues.

## 4. Immune Response to AAV

Adeno-associated virus (AAV) is a member of the Parvoviridae family [20], a non-enveloped virus that consists of a protein capsid surrounding and protecting a small, single-stranded DNA genome of approximately 4.8 kilobases (kb) [41]. AAV is the most widely used vector for ocular gene therapy due to its non-pathogenic nature [42], transduction efficiency and ability to transduce a wide variety of cell types [43]. Furthermore, recombinant AAV (rAAV) rarely integrates into the host cell genome, thereby reducing the risk of insertional mutagenesis following gene therapy [44].

Many AAV-mediated gene therapies have made a successful translation to clinical trials, and some are available as approved therapies. However, in addition to therapeutic effects and vision restoration, there have been increasing reports of adverse immune outcomes in clinical trials. In a LCA clinical trial (NCT00999609, Phase 3), mild inflammation was observed in 2 of the 21 patients who received the RPE65 gene packaged in AAV2 vectors [45]. Transient and low levels of anti-AAV2 neutralizing antibodies (NAbs) were detected in two out of three patients, one at day 14 and the other at day 90, which decreased to baseline 1-year post-injection [46]. In another LCA clinical trial (NCT02781480, Phase 1/2) delivering the RPE65 gene via the AAV2/5 vector, uveitis developed in one out of three patients in the low dose group, all three patients in the intermediate dose group and in two out of three patients in the high dose group. In NCT00749957 (Phase 1/2), three out of six patients who received a high dose of AAV showed ocular inflammation, even though the T-cell responses to capsid and the transgene product were not observed by ELISpot [47]. In another LCA trial (NCT00643747, Phase 1/2), 12 patients received either a low or high dose of AAV carrying the RPE65 gene. To investigate the long-term effect of AAV-mediated gene therapy, clinical measures of vision such as visual acuity, contrast sensitivity, color vision, and spectral sensitivities were evaluated. In this study, prednisolone was provided before and up to three years after injection. A deleterious response was observed in one participant in whom an episode of mild anterior uveitis was followed by focal pigmentary changes at the macula and a persistent reduction in visual acuity by 15 letters on the Early Treatment of Diabetic Retinopathy Study (ETDRS) chart. NAbs against AAV2 increased in this participant at week 4 and T cells specific to AAV2 capsid were also detected. Two participants had asymptomatic episodes of posterior intraocular inflammation. In one of them, the inflammation was associated with a temporary attenuation of the improvements in retinal sensitivity, while in the other, there was a transient increase in circulating NAbs against AAV2. However, angiography and fundus imaging at 12 months after therapy showed no significant change from baseline in any participant. The results showed that retinal sensitivity was improved; however, the final transgene expression in the participants from the high dose group was adversely affected by ocular inflammation or immune responses [48]. These studies confirmed that immune responses remain an important barrier to the effective application of AAV-mediated gene therapy.

These clinical reports further emphasize the need for systematic analysis and the reporting of immune outcomes from animal studies, especially in large animal models. In canines, a study designed to evaluate the efficiency and safety of AAV vectors, delivered rAAV2 carrying GFP to the retina by subretinal injection. Long-term expression was observed in photoreceptors starting from 2 weeks and persisting until 12 months. However, evidence of an intraocular inflammatory response was reported in three out of eight animals [49]. Another canine study reported improved vision in 10 out of 11 RPE65-/- animals that received AAV coding RPE65 by subretinal injections. However, 75% of the eyes that received AAV-RPE65 developed uveitis while the sham-injected and the AAV-GFP controls did not, implying that the uveitis was in response to the RPE65 gene/protein product. Additionally, there was a two-fold increase in anti-AAV antibodies in dogs that received AAV [50]. In a study aimed at evaluating the safety of AAV carrying CNGB3 for the treatment of achromatopsia, three groups of NHPs received a subretinal injection with either a vehicle or AAV-CNGB3 at a low or high dose. Anti-AAV antibodies had developed in all vector-injected animals, while no animals developed antibodies against CNGB3. Furthermore, intraocular inflammation was observed one-week post-injection and recovered with time [51]. To determine the dose required to achieve efficient transduction of photoreceptors in the NHP retina, different doses of AAV2 or AAV8 carrying GFP were delivered by subretinal injection. The animals in the study were negative for AAV-neutralizing antibodies at enrollment. High doses resulted in an increase in anti-AAV antibodies, a systemic T-cell response to the transgene and retinal damage characterized by a loss of photoreceptors [52]. To provide a detailed analysis of an innate and adaptive immune response to clinical-grade AAV8 in NHP and make a comparison with preliminary clinical data from three patients from a retinal gene therapy trial for CNGA3-based achromatopsia (NCT02610582, Phase 1/2), 34 NHPs received subretinal injections of AAV8 at low or high doses. The results revealed that three animals from the high dose groups developed mononuclear infiltration in the retina and an upregulation in the IFN-γ-mediated cytokines of the pro-inflammatory Th1 pathway four weeks after subretinal injection, as well as CD8^+^ T cells in the retina and serum antibodies. To have a better understanding of the parameters that may contribute to the immune responses induced by ocular AAV delivery, ELISA and NAb assays were performed on serum from 41 NHPs that received AAV in low, medium or high doses. Both binding antibodies and neutralizing antibodies were generated in a dose-dependent manner. There was an influence of the type of promoter, while the mode of injection did not influence the immune outcome [30]. These studies imply that in the case of large animal models, it is pertinent to take into consideration several factors such as dose, transgene cassette and vector that can influence the immune outcome (Figure 1).

## 5. Immune Response to Other Viral Vectors

As mentioned earlier, although AAV is an effective vector, it is limited by its cargo capacity, which is less than 4.7 kb [10]. While lentiviruses (LVs) can have a larger capacity that could reach up to 8 kb [11], they also integrate their genome to the target cells, which is a two-edged sword. On the one hand, this can provide long-term expression, while on the other, the integration can often occur at an oncogenic locus [53]. In a study comparing the transduction efficiency and safety of different viral vectors, GFP packaged in AAV, LV, Adenovirus (Ad) and Baculovirus (BV) was delivered intravitreally to C57BL/6OlaHsd mice. By GFP immunolabelling on retinal sections, the study concluded that LV could have a long-lasting transgene-GFP expression in the RPE, which was stronger than the expression of Ad but weaker than AAV, and the mice that received BV showed no GFP positive cells after 7 days. However, an investigation on inflammatory responses and toxicity revealed that LVs induced macrophage recruiting and anti-transgene antibody production, but not as strongly as Ads did. When comparing the F4/80 (a major macrophage marker) positive cells in retinal sections, mice that received Ad or BV showed the highest levels of F4/80 positive cells, followed by the LV group, and the lowest levels were observed in the AAV group at 3 and 7 days post-injection. Furthermore, an anti-GFP measurement performed by ELISA on serum samples revealed that Ad induced the highest levels of antibody production, followed by LV, BV and AAV [54]. The three main LV vectors used for gene therapy are human immunodeficiency virus (HIV), simian immunodeficiency virus (SIV) and equine infectious anemia virus (EIAV), which is of non-primate origin [55]. Recombinant human immunodeficiency virus type 1 (HIV-1) was the first LV to be used for ocular gene therapies. GFP packaged in HIV was delivered by subretinal injection into Fischer 344 rats and the expression of GFP was reported to have been present for at least 12 weeks without a decrease in the RPE. Several macrophages were observed surrounding the injection site, which the authors ascribed to surgical damage as a similar phenomenon was observed in the control animals [56]. In another study, HIV vectors carrying GFP were delivered by subretinal injection—a stable expression was observed in RPE, and no inflammatory cell infiltration was observed by immunostaining [57]. In a study aimed at evaluating the efficiency and safety of SIV vectors, a SIV vector was injected subretinally in adult Male Wistar rats at a low (2.5 × 10^7^ TU/mL, TU: transducing unit) or high (2.5 × 10^8^ TU/mL) dose. In the low dose group, no inflammatory reaction was observed by histology examinations, and this was accompanied by a stable long-term transgene expression which persisted over 1 year. However, mononuclear cell infiltration was found in the subretinal area in the high dose group, indicating that the dose is an essential parameter irrespective of the viral serotype [58]. Another study investigated the efficiency of five recombinant SIV vectors carrying GFP in the Royal College of Surgeons’ (RCS) rat retina. One of the tested pseudotypes expressed a high level of GFP and led to GFP-related toxicity, and the transduced cells were cleared progressively. The authors concluded that the mononuclear cell infiltration may be caused by a high level of GFP expression [59]. Thus, the nature of the transgene and its immunogenic potential must also be considered in addition to the vector type.

Adenoviruses (Ads) can offer one of the largest capacities for delivering transgenes of up to 36 kb [12]. There have been successful attempts to deliver Ad vectors to rodent retina. To restore the early cone loss in Rpe65^-^/^-^ mice, Ad expressing RPE65 was delivered to Rpe65^-^/^-^ mice and high levels of RPE65 expression were reported [60]. Delivery of Ad encoding the Mertk gene to the RPE of RCS rats (a retinal degeneration model), rescued RPE dysfunction [61]. However, one of the major drawbacks of the Ad vector is the potential immune responses it may induce, as Ad is known to cause respiratory and other kinds of infections in humans and animals [62]. In order to investigate the immune consequences of Ad, three groups of adult mice were injected with Ad carrying a lacZ reporter gene into immunocompetent, transiently immunosuppressed BALB/c mice and congenital, immunodeficient nude mice. CD4^+^ and CD8^+^ T-cell infiltration was observed in retinal sections, and anti-Ad antibodies were also measured in the immunocompetent mice. The transgene expression in BALB/c mice after the transient depletion of T-cells stayed longer than in the immunocompetent mice. The data demonstrated that the transgene expression decreased with time due to the immune response elicited by Ad, whereas when T-cells were deleted transiently, the expression could stay longer but would be eliminated once the transient depletion reagent delivery was stopped [63]. This study provided evidence of the pivotal role played by pro-inflammatory effector T-cells in the long-term expression of the transgene.

Currently, there is one active clinical trial recruiting to evaluate the safety of Ad (VCN-01) to treat refractory retinoblastoma (RTB) (NCT03284268, Phase 1). Two clinical trials have been conducted to test treatments for AMD [64] and retinoblastoma [65]. Out of the 28 patients with AMD, 25% developed transient intraocular inflammation. However, the systemic immune responses, which were tested by measuring the neutralizing antibody levels, were low and inconsistent. However, several patients showed a small increase (less than 2 log units) and one patient showed a 3 log-unit increase at 3 weeks, which decreased to baseline at week 12 [64]. The retinoblastoma cohort consisted of eight patients, five of whom developed ocular inflammation even though there was no measurable humoral or cellular immune response [65].

## 6. Immune Response to Non-Viral Vectors

Non-viral delivery methods are being increasingly explored as potential alternatives to viral vectors in order to have flexibility with respect to the size of the cargo and avoid viral components. These methods can include the direct delivery of proteins or DNA by physical or chemical methods, such as conjugation with liposomes, polymers and nanoparticles (Figure 1) [66]. However, so far, non-viral vectors have proven to be less efficient than viral vectors [67].

The simplest gene transfer system is to inject naked plasmid DNA by electroporation, ultrasound or a gene gun [67], although there is a risk of being degraded by nucleases in the serum and cleared by phagocytosis [17]. The gene gun was one of the early techniques developed for gene delivery to accessible tissues. One study reported the delivery of GFP conjugated with gold nanoparticles to the rabbit cornea. The study reported a GFP expression without the presence of inflammatory cells in the cornea [68]. To determine the efficiency and safety of naked, plasmid gene therapy to the corneal stroma and epithelium, naked DNAs expressing GFP, beta galactosidase (β-gal), vascular endothelial growth factor (VEGF), or soluble Flt-1 (s-Flt) were injected into the cornea of CD-1 mice. The transgene expression was found as early as 1 h, peaked by 24 h and decreased from then on. Inflammatory cells were observed in the histological preparation of the cornea, and vascular engorgement was observed after a high-dose injection of VEGF or GFP. These results provided proof for the possibility of a fast but transient gene expression with naked DNA delivery, especially to easily accessible tissues such as the cornea. However, even with a transient short-term expression, immune responses can be a cause for concern [69]. To achieve a more sustained gene delivery in the retina, two plasmid vector backbones—pEPI-1 and pEPito—were designed to overcome the limitations of a low transgene expression, vector loss during mitosis and gene silencing. pEPI-1 contains a scaffold/matrix attachment region (S/MAR) which is involved in DNA duplex destabilization and strand opening. pEPito was constructed by cloning the pEPI-1 plasmid replicon in a plasmid backbone containing 60% less CpG motifs and was less prone to silencing. These two plasmids were tested in an in vitro system (RPE cell line—D047) and in vivo by intravitreal injection in WT C57BL/6 mice. Both plasmids were able to transduce RPE cells, with a sustained expression observed for up to 3 months in vitro and 32 days in vivo (the last observation time-point). In vivo, the GFP expression was lower in the pEPI-1 group compared to the pEPito group, which the authors attributed to less CpG motifs in pEPito. However, other aspects of inflammation were not observed, most likely because this study used an immunosuppressive regimen [70]. The major limitation of using naked DNA is their degradation and eventual clearance by cellular systems. Indirect evidence for this was provided by a study of the cGAS-STING pathway in retinal degeneration. The cGAS-STING pathway is part of the innate immune system that detects cytosolic DNA and triggers the expression of inflammatory genes. During the process of retinal degeneration, cytosolic DNA along with other cellular debris, remain at the site of degeneration, which can activate a cGAS-STING-mediated immune response. To explore the function of cGAS-STING in oxidative stress (OS)-induced retinal degeneration, C57BL/6J mice were injected with the oxidant sodium iodate. DNA leakage was found in the retina after OS, which promoted the expression of cGAS and STING, which further promoted retinal inflammation, as an infiltration of microglial cells were observed. Besides, cGAS and STING participate in DNA clearance too. The study demonstrated that cGAS-STING would be activated when cytosolic DNA is found in the retina, resulting in DNA clearance and local inflammation, which is likely to be a barrier for naked DNA delivery [71]. The sequence of the DNA being delivered can also influence the immune response, as unmethylated CpG-DNA released in hos7t cells are recognized by TLR9. When CpG oligodeoxynucleotides (ODNs) were injected in C57BL/6 mice, neutrophilic infiltration was observed at the corneal stroma and a macrophage response induced by CpG ODNs was observed, confirming that naked DNA with CpG would induce ocular inflammation [72].

Liposomes, polymers such as polypeptides and polysaccharides, and nanoparticles are among the chemical methods currently available for non-viral gene delivery. These form a complex with DNA, drugs or proteins, and are transferred into cells [7]. To test the feasibility of transfecting retinal ganglion cells (RGC) using polyethylenimine (PEI)/DNA polyplexes, a polyplex of PEI conjugated to a plasmid expressing shRNA that was targeting melanopsin and a DsRed reporter was injected intravitreally in WT mice. The DsRed expression was observed in GCL, while the melanopsin expression decreased, providing evidence for the suitability of polyplexes for ocular gene delivery [73]. Although these polyplexes do not contain any viral material, they are still foreign materials and they may lead to the immune response by the host [74]. To test the toxicity of PEI, ex vivo human cornea explants and in vivo rabbit cornea received PEI conjugated to gold nanoparticles (PEI2-GNP). Slit lamp exams of the rabbit cornea did not show any inflammation and stained gold nanoparticles were detected in the stroma of rabbit corneas. TUNNEL staining was carried out to assess cell death on explants that showed positive cells at 72 h, but the numbers were not significantly different from the negative controls (Sharma et al., 2011). Another study compared the toxicity induced by three different materials—Chitosan (CHI), poly{[(cholesteryl oxocarbonylamido ethyl) methyl bis(ethylene) ammonium iodide] ethyl phosphate} (PCEP) and magnetic nanoparticles (MNP). GFP packaged in CHI, PCEP and MNP was delivered intravitreally or subretinally to rabbits, and the transgene expression and immune response were analyzed at day 7. Inflammatory cell infiltration and retinal degeneration were observed in the rabbits that received the CHI vector but not the others. Neither PCEP nor MNP induced inflammatory cell infiltration in both ways of injection, while retinal degeneration was observed in 15% of the eyes that received PCEP subretinally. Further, immunolabelling revealed that neither CHI nor PCEP were able to transfect ocular cells efficiently. These results further reiterate that non-viral vectors not only lack efficiency for ocular delivery applications but can also induce immune responses [75]. A study analyzed the efficiency and safety of lipoplexes carrying the ribonucleoprotein (RNP)—SpCas9 and a sgRNA targeting *Vegfa*—by subretinal delivery to WT C57BL/6J mice. The GFP expression and indel formation frequency increased with the quantity injected, reached a peak and then started to decrease if higher doses were used. Increased autofluorescence was found in RPE cells as a sign of toxicity following the delivery of lipoplexes containing high amounts of RNP [76]. Interestingly, to evaluate the effects of gold nanoparticles (GNP) in endotoxin-induced uveitis in rats, GNPs were delivered to Wistar rats with lipopolysaccharide (LPS). The rats that received LPS only showed a high expression of TNF-α, TLPR4 and NFκB, meaning that inflammation was induced by LPS, while the rats that received LPS and GNP had a significantly lower expression. These findings interestingly suggest that topical GNP decreases intraocular inflammation and oxidative damage by interfering with the TLR4–NFκB pathway [77]. Even though one of the early motivations for developing non-viral vectors was to avoid viral components to have lower immunogenicity [78,79], these studies highlight that depending on the non-viral vectors used, the immune consequence could be severe, as is in the case of chitosan, or could be protective, as shown by the GNP study.

A major drawback of unpackaged systems is their susceptibility to cellular degradation mechanisms, and a major drawback of lipid-based packaging is their large size and lack of cell-specific targeting. To circumvent these issues, there has been a recent shift towards engineering delivery systems that are synthetic but based on design principles of viral capsids called Virus-like particles (VLPs) (Figure 1). VLPs are nanoscale structures made up of one or more different molecules with the ability to self-assemble, mimicking the form and size of a virus particle, but lacking the viral genetic material and being non-infectious [80]. VLPs can be used as carriers to deliver small molecules, genes and nucleic acids, or peptides and proteins in cancer therapies, immunotherapies, vaccines and gene therapies [81]. One of the most exploited applications of VLPs is in vaccine development, wherein they are modified to present epitopes that elicit specific T-cell and humoral responses [82]. In a recent study, the authors engineered DNA-free VLPs to delivery adenine base editor (ABE) as a RNP complex to the different organs of mice, such as the brain, liver and eyes. They showed that in primary human and mouse fibroblasts, VLP-mediated base editing could be achieved with more than 95% efficiency. Further, in vivo in the central nervous system, the authors induced a silent mutation in the mouse *Dnmt1* gene by neonatal cerebroventricular injections and achieved 53% editing in the cortex and 55% editing in the midbrain. In the liver of C67BL/6J mice, they targeted Pcsk9, a therapeutically relevant gene involved in cholesterol homeostasis, and showed 63% editing. VLPs were injected subretinally in *rd12* mouse, which is a LCA model harboring a mutation in the *Rep65* gene. Five weeks post-injection, they reported 21% editing in the RPE cells and the rescue of visual function, as measured by ERG. This study provided compelling evidence for the use of VLPs for in vitro and in vivo applications. Although analyses of immune responses were not performed, they reported low toxicity based on the data from liver delivery [83]. Another study developed murine-leukemia-based VLPs carrying the Cas9-gRNA RNP complex (named nanoblades) and showed 67% editing efficiency in human-induced pluripotent stem cells (hiPSCs) targeting the *EMX1* gene. They further tested nanoblades on mouse zygotes to demonstrate that they can be used to generate transgenic animals. Here, the editing efficiency showed large variability, ranging from 11% to 78%. In the mouse liver, they were able to show 7–13% editing efficiency [84]. A recent study developed a modular delivery system called selective endogenous encapsidation for cellular delivery (SEND). Eukaryotic genomes contain genes from integrating viruses and mobile genetic elements that include homologs of the capsid protein (known as Gag), long terminal repeat (LTR) retrotransposons and retroviruses. They identified a LTR retrotransposon called PEG10 that caused vesicular secretion of its own mRNA, and engineered the PEG10 to carry mRNA cargo of a gene-of-interest by flanking it with Peg10’s untranslated regions (much similar to the ITRs of rAAV). The SEND delivery, using the modified PEG10, was adapted for application in both mouse and human cell lines, and they showed up to 40% editing of the *VEGFA* gene in HEK cells. Immune responses have not been evaluated, but SEND may have reduced immunogenicity compared to currently available viral vectors due to its use of endogenous human proteins [85].

## 7. Immunosuppressive Strategies

It is becoming increasingly evident that both viral and non-viral deliveries to the ocular space are likely to cause an immune response—this can result in inflammation, clearance of transduced cells and deterioration of vision. Thus, considerable efforts are being focused on devising strategies to prevent, manage or overcome these adverse responses. These strategies can be broadly categorized as vector-oriented and immune response-oriented approaches (Figure 2).

### 7.1. Vector-Oriented Strategies

Dose optimization: Several studies have shown that the dose of the vector administered influences the immune outcome. An increase in dose implies an increase in vector, transgene and promoter—each of which has independently and cumulatively shown to affect immune responses [30,52,86,87]. Thus, the optimization of these parameters can potentially modulate the immune response (Figure 2).

Capsid design: The vector capsid can be engineered by directed evolution to make novel variants that can escape or resist immune response (Figure 2). Further, these methods can also create more efficient AAVs which allow us to reduce the dose, and in turn, reduce the immune responses. A library of AAV2 variants was generated by error-prone PCR, and the variants were screened for their ability to transduce cells in the presence of serum antibodies. The best-performing variants selected from this screen were reported to evade antibodies and enhance the gene expression [88]. Another study used a structure-guided approach, wherein cryoelectron microscopy and 3D image reconstruction were used to analyze the AAV-antibody complexes to find the antigenic footprint on the AAV capsid. Variants developed based on this analysis required 2- to 16-fold higher antibodies for neutralization compared to their parental serotype [89]. Another study employed a DNA shuffling–based approach to generate variants and screen them to find a less immunogenic variant. A novel variant selected from this screen—chimeric-1829—showed higher transduction efficiency and low cross-reactivity with the pre-existing antibodies against other serotypes, indicating that the immune response may not recognize chimeric-1829 [90].

Capsid decoys: To enhance the safety of the AAV gene therapy, a study mutated the receptor binding site of AAV2 and generated empty capsids that can adsorb antibodies but cannot enter a target cell. When AAV8-hFIX with mutated AAV8 decoys were injected in WT C57BL/6 mice immunized with intravenous immunoglobulin (IVIg), the transgene expression increased with the amount of AAV empty capsid [91]. However, this technique has not been used in the eye, and given the small ocular space and dosage restriction, using large amounts of empty decoy capsids may not be beneficial.

Helper-dependent Adenovirus (HdAd): To decrease the influence of immune responses induced by Ad, a helper-dependent vector system was developed by removing most viral sequences, except inverted terminal repeats (ITR) and the packaging signal [12]. A comparison between HdAd and first-generation Ad (FGAd) was made by delivering GFP intravitreally in a rat model of oxygen-induced retinopathy. HdAd promoted a long-lasting (up to one year) transgene expression in the retina, whereas the FGAd-mediated expression decreased after two months due to immune response-related clearance [92]. Another study delivered HdAd-encoding centrosomal protein 290 (CEP290)—a causative gene associated with LCA—to CD-1 mice to test the transduction efficiency and toxicity. A long-term gene expression without a decrease was reported [93]. Thus, the application of HdAd vectors in ocular gene therapy can potentially improve the efficiency and safety.

CpG motif deletion and TLR inhibition: CpG islands are clustered regions rich in CG motifs that are common in AAV vector sequences and promoter sequences. An immune response results from the interaction of these CpG motifs with TLR9 receptors. This CpG–TLR9 interaction was demonstrated by inducing an immune response in WT mice by an immunogenic AAV variant, and this response was attenuated in TLR9-/- mice [94]. The depletion of CpG sequences in the viral genome helped reduce immune responses (Figure 2) [72,95]. A study showed that short TLR9 inhibitory sequences called TLRi (approximately 12 to 24 nucleotides) could be incorporated into the viral genome to block TLR9 activation. The TLRi-containing vectors elicited reduced immune responses and enhanced the gene expression in the retinas of mouse and pig models [94].

Masking Cas9 epitopes: In order to make Cas9 evade immune responses, a study identified two immunodominant SpCas9 T cell epitopes for HLA-A*02:01 using an enhanced prediction algorithm. This prediction incorporated T-cell receptor contact residue hydrophobicity and HLA binding, and evaluated them by T-cell assays using healthy donor PBMCs. When SpCas9 with mutations in the selected epitopes was delivered to healthy donor B-cell APCs, the measured T-cell response had significantly decreased. This finding demonstrated that Cas9 proteins can be modified to eliminate immunodominant epitopes through targeted mutations while preserving its function and specificity (Figure 2) [96].

Cas9 orthologs: Another method would be to identify and develop new orthologs from non-pathogenic bacteria or bacteria from extreme habitats that are not known to have any prior exposure to humans (Figure 2). One such novel ortholog of Cas9 protein was derived from the thermophilic bacterium, Geobacillus stearothermophilus (GeoCas9), which edited mammalian genomes effectively. In GFP-expressing HEK293T cells, GeoCas9 showed editing efficiency comparable to SpCas9 when used for the disruption of GFP function [97]. As Geobacillus stearothermophilus thrive in harsh environments such as hot spring cool soils [98], humans may not have as a high prevalence of antibodies against GeoCas9 as SpCas9.

### 7.2. Immune Response-Oriented Strategies

Immunosuppression: An approach to reduce the inflammation and immune response related to gene therapy is to use non-specific immunosuppressive agents (Figure 2). In a study aimed at evaluating the AAV-Cas9 immune responses in mice, providing FK506 daily at 5 mg/kg reduced the number of activated CD4^+^ and CD8^+^ T-cells that were specific to Cas9 [40]. In another study involving the delivery of AAV-Cas9 in DMD canines, all the animals injected with Cas9 were administered with a high-dose of prednisolone (orally at 4 mg/kg once a day for 3 days before AAV injection and continued for 7 days after AAV injection). Both humoral and cellular immune responses were still observed, and reduced transduction efficiency was reported [5]. However, in this study, it is difficult to assess the role played by immunosuppression due to a lack of groups without prednisolone. Although corticosteroids are routinely used in clinical trials for immunosuppression, there have not been any systematic comparative studies evaluating their effects. Even in large animal models and translational studies, immunosuppression is used but is provided as per the recommendation of the veterinarian, and not always systematically evaluated or even reported.

Tlr9 Receptor blocking: Surface receptors called TLRs are displayed on immune cells, and TLR9 normally senses DNA from pathogenic viruses that contain unmethylated CpG motifs. CpG binding to TLR9 promotes its dimerization and activates TLR9 signaling. This leads to innate immune responses eventually recruiting other immune cells to the site of infection and further triggers adaptive immune responses (Figure 2) [99]. When cationic lipid conjugated with plasmid DNA was injected into WT and TLR9-/- mice intravenously, serum cytokine levels were significantly higher in WT compared to the TLR9-/- mice [95]. Another study compared immunological responses in WT and Tlr9-deficient mice that received an immunogenic AAV variant (AAVrh32.33). In Tlr9-deficient mice, IFN-γ T cell responses toward capsid and transgene were suppressed, resulting in a minimal cellular infiltrate and stable transgene expression [94].

Plasmapheresis: The removal of all blood plasma is known as plasmapheresis, and it is carried out in order to deplete the pre-existing antibodies as well as eliminate the possibility of an immune response. This has been demonstrated on patients—non-human primates [100,101] and rats [102]—who received systemic injections of gene therapy [103]. There have been no studies demonstrating the effectiveness of plasmapheresis in ocular studies; considering that this procedure is very invasive and drastic, its application in the eye will probably have more drawbacks than merits.

Endopeptidase treatment: Imlifidase (IdeS) is an endopeptidase that can cause enzymatic degradation of circulating IgGs. A study passively immunized mice with serum IgGs, followed by IdeS treatment and AAV injection, and showed decreased anti-AAV antibodies and an improved gene transfer. IdeS administration was shown to be safe and efficient in non-human primates too. Further in vitro testing of human plasma samples collected from gene therapy trial patients resulted in reduced anti-AAV antibody levels after Ides treatment (Figure 2) [104].

ACAID and SRAII mechanisms: The immune-privilege accorded to the eye is partly due to the presence of special mechanisms such as anterior chamber-associated immune deviation (ACAID) and subretinal-associated immune inhibition (SRAII). In ACAID, an antigen injected into the anterior chamber of the eye is taken up and processed by antigen-presenting cells (APCs) that migrate to the thymus and the spleen. These F4/80^+^ CD11b^+^ ocular APCs further generate immunomodulatory cells such as CD8^+^ or CD4^+^ regulatory T cells (Tregs) which then spread through the body and induce antigen-specific immune deviation [105]. SRAII is a similar mechanism demonstrated in the subretinal space [106]. Thus, an initial activation of these mechanisms could potentially modulate the immune response to subsequent gene therapy (Figure 2).

## 8. Current Challenges and Future Perspectives

Gene therapy by gene replacement and gene editing shows great potential in treating IRDs. However, the therapeutic components such as the vector and/or the transgene can induce both innate and adaptive immune responses. In several cases, patients require repeat injections, thus, increasing the potential for immune responses. Even when different AAV serotypes are used, the cross-reactivity across serotypes remains a major challenge. A recent study attempted to overcome this challenge by using combinations of AAV and Cas9 orthologs that had reduced cross-reactivity and distinct immune response profiles, such that they caused a lower immune response upon repeat injections [107]. In CRISPR–Cas9 system-based therapies, pre-existing immunity has been reported against SaCas9 and SpCas9, as discussed before. Furthermore, the guide RNA also poses a risk of immune response activation, which in turn affects the editing efficiency. A study reported high-frequency off-target mutagenesis in transduced human cell lines after SpCas9 delivery [108]. These unwanted mutations may cause a loss of gene function, can be oncogenic or present danger signals that could induce an immune response [109]. To reduce off-targets, the guide RNA design needs to be optimized, taking into consideration the GC content, length and chemical modifications [109,110]. The Tlr9 inhibition (Tlr9i) sequences added to the transgene packaged into AAV vectors vastly reduced the immune responses. This strategy can potentially be combined with the Crispr-Cas9 system to probably achieve a similar inhibition. In order to improve the safety and efficiency of the CRISPR–Cas9 system-based therapies, new orthologs of the Cas9 protein with low immunogenicity could be developed from microorganisms to which humans are not commonly exposed [97,111]. Cas9 is of bacterial origin, and hence, is expected to cause an immune response [112]. Human orthologs of Cas9 could be advantageous, but these do not exist. Hence, a possibility is to synthetically replace parts of the bacterial protein sequences (specifically the immunogenic ones) with sequences of human origin.

Another major challenge is the reporting (or the lack thereof) of immunosuppressive regimens which are routinely administered before and after the gene therapy intervention. Immunosuppression is often administered as per a physician’s prescription; these are rarely reported in animal studies and are inadequately documented in most human studies. Hence, a more systematic and consistent analysis of amounts, types and durations of immunosuppression can prove beneficial for designing future studies.

As a next step, immune deviation mechanisms such as ACAID or SRAII can be explored for the inhibition of immune responses. Furthermore, a study showed that CD4^+^CD25^+^ regulatory T cells (Tregs) could inhibit immune responses to transgene in mice [113]. Currently, human antigen-specific Treg cells are not approved for human use in vivo, but this strategy might be approved in the near future. A future perspective with respect to immunosuppression would be to use a combination of approaches, such as activating the ACAID mechanism prior to gene therapy and administering immunosuppressive agents locally post-therapy. A major challenge has always been that immune responses are highly individualistic, hence, drawing concrete conclusions about immune responses from a small cohort of patients has been difficult. However, as more therapies are tested, the cohorts become larger and more immune response-related data become available, and the prediction of immune responses is likely to become more reliable in the near future. The next step would be to aim to personalize the immunosuppression combination regimen, taking into account each person’s immunological status.

## 9. Conclusions

Viral vectors, such as AAVs, which are non-pathogenic and have translated well into clinical applications, are being increasingly implicated in adverse immune consequences. However, all the initial developments in AAV-based therapies have largely ignored analyzing and/or reporting these immune consequences. On the other hand, Ads have always been known to be immunogenic, and hence, have been applied with caution, with more Ad studies reporting immune consequences. With non-viral vectors, there have been attempts to develop the vectors and simultaneously pay attention to the immune responses that these novel vectors can elicit. Additionally, this is a trend that needs to be continued so that when the non-viral vectors are ready to transition into clinical applications, more consistent data are available regarding the potential immune responses. CRISPR–Cas9 system-based therapies are being developed at a fast pace, and in the near future, a lot of these therapies are likely to make their way into clinics. As Cas9 is a protein involved in the bacterial defense mechanism, it was believed that there will be low immunity in humans against Cas9. However, some recent studies have shown a high prevalence of anti-Cas9 antibodies in humans. These early studies have already prepared us to expect immune responses to CRISPR–Cas9-based therapies in clinical settings. In anticipation, there are ongoing efforts to develop techniques—such as Cas9 epitope masking or the identification of novel Cas proteins, such as GeoCas9—to circumvent this problem of anti-Cas9 pre-existing adaptive immunity. Furthermore, when referring to immune responses, it is important to make a distinction between acceptable immune responses and adverse immune responses. When any foreign substance or intervention is applied to a healthy body, an immune response can occur, which indicates a normal and expected function. However, when this immune response is aggravated, spreads outside the area of the intervention and cannot be managed, it can be referred to as an adverse immune response. In the context of gene therapy applications in the eye, immune response resulting in uveitis is a relatively common occurrence post-intervention and can often be managed by prescription medicines or immunosuppression. However, this becomes a cause for concern when the inflammation does not subside over time, leading to the clearance of transduced cells and resulting in reduced visual acuity. Thus, while designing and optimizing different parameters of the gene therapy, one must not only consider the type of immune response that may occur, but more importantly, if these responses will adversely affect the therapeutic outcome. All of the studies reported so far measure or observe different components of the innate and adaptive immune system, and report the presence or absence of the immune response. A combination of immunomonitoring tools should be used to better understand the different immunological parameters. One aspect that is missing so far is the mechanistic insights into how exactly the responses occur at cellular or molecular levels. Such studies can provide crucial information towards devising new strategies to evade and manage these responses and make ocular gene therapies safer and more efficient.

## Figures and Tables

**Figure 1 pharmaceutics-14-01973-f001:**
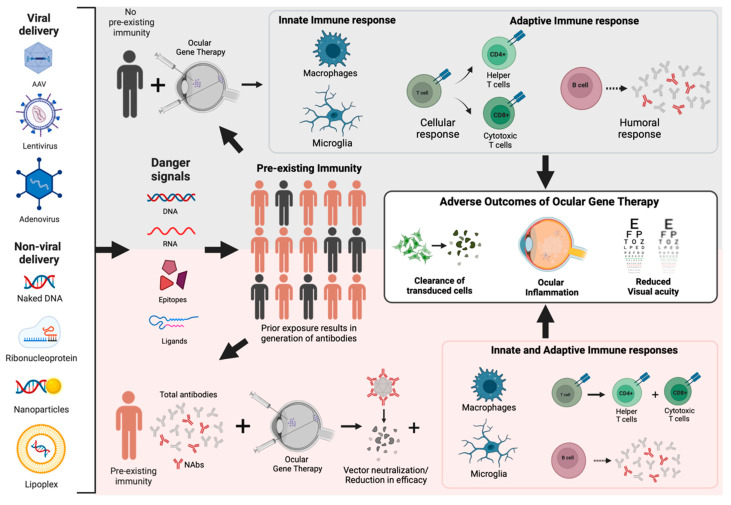
Schematic representation of immune responses to ocular gene therapy. Some components of viral and non-viral delivery methods present danger signals to the immune system. Exposure to these components prior to gene therapy can result in pre-existing immunity. Individuals without prior exposure can activate innate and adaptive mechanisms after receiving ocular gene therapy. Individuals with pre-existing immunity can generate neutralizing antibodies (NAbs) that eliminate the vector and further trigger innate and adaptive responses. Together, these responses can result in an adverse outcome of ocular gene therapy, such as clearance of transduced cells, inflammation and reduction in visual acuity.

**Figure 2 pharmaceutics-14-01973-f002:**
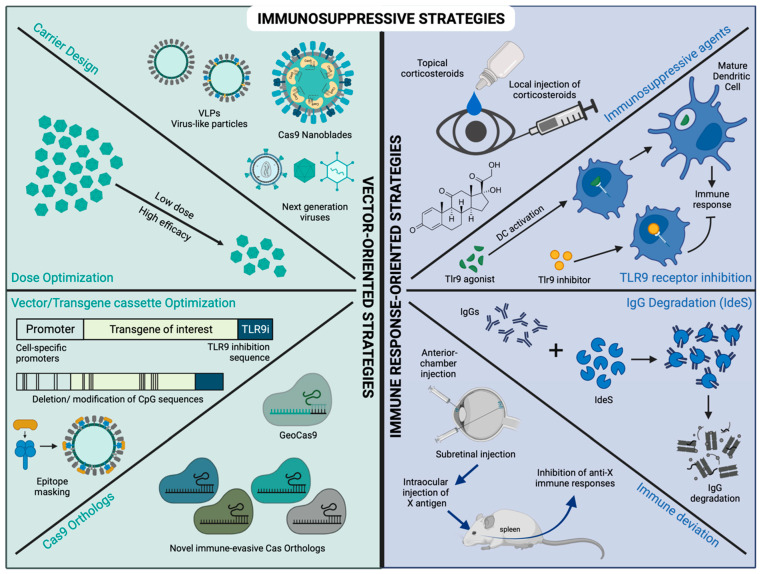
Schematic representation of immunosuppressive strategies. Strategies to suppress or control immune responses to gene therapy can be broadly categorized as vector-oriented or immune-response-oriented strategies. Vector-oriented strategies focus on design and dose of the carrier and the elements packaged within. Novel carriers are being designed with immune-evading properties such as the next generation of viruses as well as non-viral carriers such as virus-like particles (VLPs) and nanoblades. A higher dose poses a higher risk of exposure to danger signals; hence, the dose needs to be optimized to achieve the highest therapeutic efficacy with the lowest dose possible. Cell-specific promoters are more specific and less immunogenic than ubiquitous promoters. An addition of a TLR9 inhibition sequence (Tlr9i), deletion of CpG-rich regions in the transgene cassette and masking of TLR9 binding epitopes on the capsid have been shown to reduce TLR9-mediated immune responses. Novel Cas9 orthologs, to which a prior exposure has not occurred, may be less immunogenic. Immune-response-oriented strategies focus on avoiding or reducing the immune responses. An immunosuppressive regimen consisting of corticosteroids can be provided as topical eye drops or ocular injections before and after the gene therapy. TLR9 inhibitors can be used to prevent the activation of dendritic cells, thereby preventing cellular immune responses. Enzymatic cleavage of pre-existing anti-vector antibodies can be achieved by using enzymes such as IdeS. Immune deviation can be achieved by prior exposure to antigens by injection into the anterior chamber or subretinal space.

## Data Availability

This statement is not relevant for the present review as no experimental data was generated or presented.

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
