# Peer review of "Immune Responses to Gene Editing by Viral and Non-Viral Delivery Vectors Used in Retinal Gene Therapy"

_pharmaceutics, 2022, doi:10.3390/pharmaceutics14091973_

Round 1

Reviewer 1 Report

The manuscript entitled “Immune responses to gene editing by viral and non-viral delivery vectors used in retinal gene therapy”, by Ren et al have attempted evaluate and discuss both innate and adaptive immune responses to gene editing both with non-viral and viral delivery in the ocular space. The review will need major work to properly convey the topic of interest of the authors to the readers. Some suggestions are as follows:

1. The article needs more images (3-4) to convey the content and making it more interesting to the readers.

2. Author should add "current challenges and future perspectives" section in the manuscript.

3. Author should also compiled important information in the form of tables.

4. Manuscript should be revised for grammatical and typographical errors.

5. Author should also add his/her opinion in this particular area.

Author Response

The manuscript entitled “Immune responses to gene editing by viral and non-viral delivery vectors used in retinal gene therapy”, by Ren et al have attempted evaluate and discuss both innate and adaptive immune responses to gene editing both with non-viral and viral delivery in the ocular space. The review will need major work to properly convey the topic of interest of the authors to the readers. Some suggestions are as follows:

1. The article needs more images (3-4) to convey the content and making it more interesting to the readers.

We agree with the reviewer that a schematic representation of the content can make it more interesting for the reader. Hence, we have added another figure to the manuscript showing an overview of all the immunosuppressive strategies (Refer Figure 2, Page 25). We believe that the two figures in the current manuscript are very comprehensive and encompass all the key concepts from our review.

 2. Author should add "current challenges and future perspectives" section in the manuscript.

As per the reviewer’s suggestion, we have now added a section ‘Current challenges and future perspectives’ Lines 595 to 634

 3. Author should also compiled important information in the form of tables.

While we agree that tables can be useful for display of categorical data, we do not believe that we have any information in the review that can be categorized and presented in a tabular format.

 4. Manuscript should be revised for grammatical and typographical errors.

The manuscript has been re-checked for errors.

 5. Author should also add his/her opinion in this particular area.

For the entire manuscript we have aimed to make a compilation of studies by summarizing existing studies for each topic that we have addressed without adding personal opinions to each section. We have expressed all our thoughts and opinions in the sections ‘Current challenges and future perspectives’ and ‘Conclusion’.

Reviewer 2 Report

With great pleasure I have red this review. This manuscript describes immune responses to viral and non-viral vectors used in retinal gene therapy. The manuscript gives the deep insight in the questions described. The article describes many cases of immune response on different kinds of vectors. The literature analysis is perfect with using of more than 100 references. As for English, I am not native speaker, for me English is acceptable. This review will be very useful for wide range of scientists studying viral and non-viral vectors for gene therapy. I think it must be published, however, after small corrections.

Minor:

1) One figure is not enough for such good review. Please, add more figures to illustrate the text.

2) In clinical trials descriptions, please, add the indications to phase of trial: for example, NCT02781480, phase 2, etc. (it is better for understanding).

Author Response

With great pleasure I have red this review. This manuscript describes immune responses to viral and non-viral vectors used in retinal gene therapy. The manuscript gives the deep insight in the questions described. The article describes many cases of immune response on different kinds of vectors. The literature analysis is perfect with using of more than 100 references. As for English, I am not native speaker, for me English is acceptable. This review will be very useful for wide range of scientists studying viral and non-viral vectors for gene therapy. I think it must be published, however, after small corrections.

The authors would like to thank the reviewer for taking the time to review our manuscript. Their constructive comments have helped us improve the content of this review. 

Minor:

1) One figure is not enough for such good review. Please, add more figures to illustrate the text.

We agree with the reviewer that a schematic representation of the content can make it more interesting for the reader. Hence, we have added another figure to the manuscript showing an overview of all the immunosuppressive strategies (Refer Figure 2, Page 25). We believe that the two figures in the current manuscript are very comprehensive and encompass all the key concepts from our review.

2) In clinical trials descriptions, please, add the indications to phase of trial: for example, NCT02781480, phase 2, etc. (it is better for understanding).

As per the reviewer’s suggestion, these changes have now been included in the manuscript – Lines 245, 248, 251, 253, 290 and 356

Round 2

Reviewer 1 Report

The manuscript can be accepted in the current form. The authors have made suitable changes, which improved the quality of the manuscript.